# Experimental Investigation on Post-Peak Permeability Evolution Law of Saturated Sandstone under Various Cyclic Loading–Unloading and Confining Pressure

**Liang Chen** [1,2,3], **Dongsheng Zhang** [1,2], **Wei Zhang** [1,2,*], **Jingna Guo** [4], **Nan Yao** [3], **Gangwei Fan** [1,2], **Shizhong Zhang** [2], **Xufeng Wang** [1,2] **and Peng Wu** [5]

1 State Key Laboratory of Coal Resources and Safe Mining, China University of Mining and Technology, Xuzhou 221116, China; chenliang_cumt@126.com (L.C.); dshzhang123@126.com (D.Z.); fangw@cumt.edu.cn (G.F.); wangxufeng@cumt.edu.cn (X.W.)
2 School of Mines, China University of Mining and Technology, Xuzhou 221116, China; zhangshizhong@cumt.edu.cn
3 Hubei Key Laboratory for Efficient Utilization and Agglomeration of Metallurgic Mineral Resources, Wuhan 430081, China; yaonan@wust.edu.cn
4 School of Mechanics and Civil Engineering, China University of Mining and Technology, Xuzhou 221116, China; tb18030001b4@cumt.edu.cn
5 State Key Laboratory for Geomechanics and Deep Underground Engineering, China University of Mining and Technology, Xuzhou 221116, China; pengw@cumt.edu.cn
* Correspondence: zhangwei@cumt.edu.cn

**Abstract:** The permeability evolution law of saturated rock under cyclic loading–unloading after shear yield is an important basis for revealing the water resistance performance and water inrush risk of overlying rock under multiple mining conditions. In this paper, the influence of the confining pressure, the cyclic loading–unloading times (CLT), and the volumetric strain on the post-peak permeability of saturated sandstone was studied by carrying out a post-peak permeability experiment. Based on SEM images and an improved simulated annealing algorithm, the 3D internal structure characteristics of sandstone samples before and after the experiment were reconstructed. The influences of the confining pressure on pore diameter, effective porosity, connectivity, seepage path length, and tortuosity of the sandstone before and after the experiment are discussed. Research results indicated that (1) In the post-peak cyclic loading–unloading stage, the volumetric strain is negatively correlated with permeability. At the unloading and initial loading stage, the volumetric strain showed a gradually decreasing trend as the specimen was slowly compressed. However, at the middle and final loading stages, the volumetric strain curve shifted to the left and showed a decreasing trend, resulting in an obvious increase in permeability. (2) The influence of CLT on *k* is closely related to the confining pressure level. When the confining pressure changed from 4 MPa to 12 MPa, the volumetric strain–average stress hysteretic curve shifted to the left in turn and the corresponding permeability gradually increased. When the confining pressure increased to 16 MPa and 20 MPa, the volumetric strain–average stress hysteretic curve shifted to the right in turn and the corresponding permeability showed a decreasing trend. No matter what the value of CLT, the magnitude of sandstone permeability gradually decreased and the decreasing trend became flat as the confining pressure increased, especially for $\sigma_3$ = 16 MPa and 20 MPa. (3) No matter what value of the confining pressure, the hysteresis area of the first cycle was larger than that of last three cycles, indicating that the plastic deformation generated in the first cycle was larger than that generated in the last three cycles and the recovery rate of the permeability increased with an increase of CLT. (4) As the confining pressure gradually increased, the pore diameter, effective porosity, and connectivity all approximately showed a linear decrease due to more easily compacted pores and cracks under high confining pressure, lower connectivity, and permeability, while the length and tortuosity of the seepage path increased nonlinearly, roughly due to a more significant shear failure phenomenon where the seepage path became more tortuous, that is, the greater the tortuosity, the longer the seepage path. The research results can provide an important theoretical basis for water resistance performance and water inrush risk assessment of overlying aquifer under the influence of mining stress.

**Keywords:** post-peak permeability; 3D reconstruction; cyclic loading–unloading; simulated annealing algorithm; non-Darcy flow

## 1. Introduction

Water inrush is one of the major hazards in coal mining safety production. To prevent and control water inrush disasters, scholars have carried out a large number of studies on the mechanism of water inrush in coal mines for more than half a century [1,2]. Among them, the seepage instability theory is one of the more rapid and widely used theoretical models in recent years. It purports that the seepage system would undergo structural instability when the initial values of surrounding rock permeability and boundary pressure meet certain conditions, where water inrush is one of the important manifestations of seepage instability of the overburdened structure [3]. Practice showed that seepage instability can only occur under the premise of overburden fracture. Therefore, it is of great significance to study the permeability variation after overburden fracture to reveal the water inrush mechanism of the working face roof in mining.

Rock permeability is closely related to its stress state. Some scholars discussed the permeability variation in the whole stress–strain process, and obtained general rules of permeability variation as follows: At the initial stage of loading, the permeability decreased slightly with the increase of stress (primary micro-fracture closure); as axial strain increased, new micro-cracks and original pores were generated in the rock and permeability increased slightly; with the continuous increase of axial strain, a perforated water flowing fracture gradually formed inside the rock, permeability increased sharply, and the obtained peak permeability lagged behind the peak stress [4–9]. In addition, some scholars discussed the influence of cyclic loading–unloading of post-peak axial stress on rock permeability and concluded that permeability was proportional to volume strain [10–13]. However, the above scholars only discussed the influence of post-peak cyclic loading–unloading on rock permeability, and the permeability variation with the increase of post-peak cyclic times was not involved, that is, the influence of rock yield and plastic flow on permeability was not considered. In the process of coal seam mining, the destruction of the overlying aquifer needs to go through a complex deformation and stress adjustment process under the influence of mining stress. During this destruction process, the plastic zone of the overlying aquifer expands continuously, and plastic flow occurs at each point in the plastic zone. The deformation of the overlying aquifer causes changes in porosity and permeability. Therefore, studying the influence of plastic flow on rock permeability is the basis for revealing the mechanism of water inrush.

In fact, the load borne by the rock is the only external condition causing the permeability variation which is essentially the result of pore structure change and crack propagation in the rock. The above studies mainly focused on the relationship between rock stress, strain, and rock permeability, but ignored influences of internal pore structure and fracture on rock permeability from the micro-perspective. Some scholars compared and analyzed micro-pore structures, fracture distribution and pattern types before and after a permeability experiment with SEM and revealed reasons for permeability variation before and after the experiment from a micro-perspective [14–17]. However, SEM has certain limitations, which can only characterize 2D pore and fracture structures of the sample and cannot directly observe 3D spatial structures of the sample. Some scholars have carried out 3D scanning experiments through CT to obtain internal 3D pore structures [18–21]. However, the high cost of CT technology and the large-scale samples led to low resolution of images. The corresponding reconstructed 3D model cannot capture the microscopic pore and fracture structures inside the rock. Some scholars also established a 3D model of pores and fractures by using a numerical calculation reconstruction method, such as the Gaussian random field method [22–24], the simulated annealing method [25–27], or the multi-point statistical method [28–31]. However, the most widely used method was the simulated annealing

method, which was based on 2D-SEM images using the random reconstruction method for numerical simulation, in order to obtain the 3D reconstruction image of the sample. Owing to the calculation of each pixel, the traditional simulated annealing method was very slow. In this paper, the calculation method of statistical function increment was improved, which can quickly and accurately perform 3D reconstruction of 2D SEM images [32,33].

Moreover, most scholars have mainly calculated the permeability through Darcy's law. The non-linear characteristics of fluid flow in rock are more obvious due to the non-uniform geometry of pores and fractures in low permeability porous media and the complex interaction mechanism between fluid and solid interface, with the result that the non-Darcy law was more applicable to the calculation of the overburden permeability after fracture. In summary, this paper studied rock permeability variation under plastic flow by carrying out rock permeability experiments under different post-peak loading–unloading paths and confining pressures. Then, an optimized simulated annealing algorithm was used to reconstruct the SEM microscopic images of rock samples before and after the experiment. The spatial distribution characteristics of pores and cracks inside the rocks under different confining pressures were analyzed, and the rock permeability variation with confining pressure under plastic flow was revealed from the microscopic point of view.

## 2. Experiment Materials and Methods

### 2.1. Sample Properties and Preparation

As shown in Figure 1, the sample was taken from the sandstone roof of the Yushuwan coal mine of Shaanxi province in China. Its average thickness was about 11.85 m. The lithology was mainly fine-grained sandstone, composed of quartz and feldspar, with good sorting, argillaceous cementation, local calcareous cementation, including small, staggered bedding, slow wave bedding and blade fossils, and fractures were developed. The main purpose of the experiment was to understand the post-peak permeability variation of sandstone in the Yushuwan coal mine with cyclic loading–unloading times (CLT) and stress levels. First, the rock sample was processed into a standard sample with a size of 50 mm $\times$ 100 mm, and then the side and upper-lower end faces were polished and smoothed by sandpaper, so that the diameter and height errors of the processed sample were less than 0.02 mm and 0.05 mm, respectively. Finally, the sample density and longitudinal wave velocity were measured, and the average values were 2308 kg/m$^3$ and 2334 m/s, respectively. The samples with large difference in the experimental results were removed. The scanning electron microscope (SEM) experiment and mercury injection experiment were carried out on sandstone to understand the basic properties of sandstone permeability, such as the initial fracture structure characteristics and initial porosity. Figure 2 shows the micro-structure images of sandstone at 1000$\times$ magnification with SEM. It can be seen from Figure 2 that the sandstone in the initial state contained more micro-cracks, and the micro-cracks were circuitous and bending without obvious strike. Moreover, the crack intersection phenomenon was obvious with the loose rock structure. In addition, the initial effective porosity of the roof sandstone was about 30.36% and the initial permeability was $3.53 \times 10^{-17}$ m$^2$.

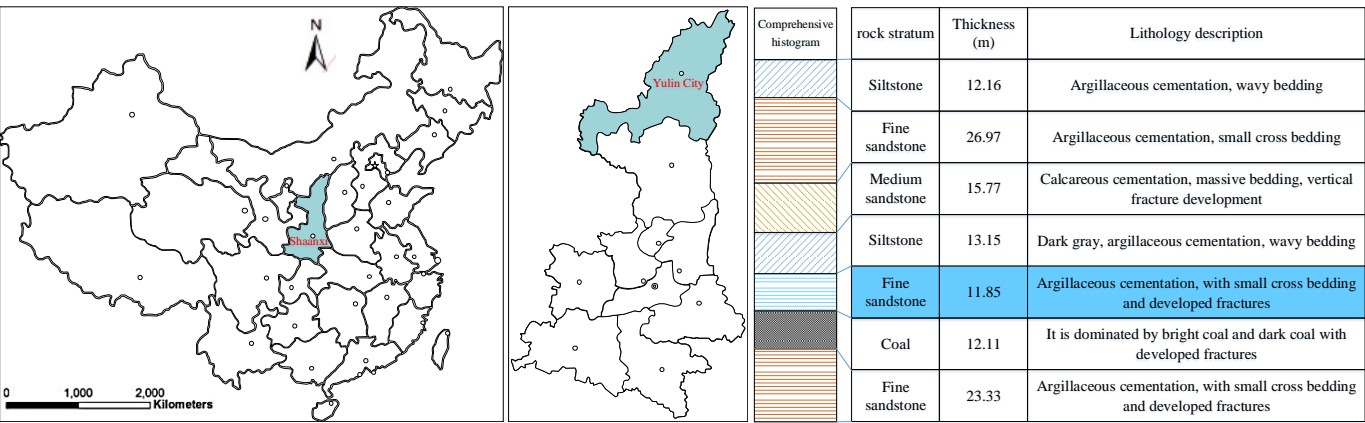

| Comprehensive histogram | rock stratum | Thickness (m) | Lithology description |
|---|---|---|---|
| | Siltstone | 12.16 | Argillaceous cementation, wavy bedding |
| | Fine sandstone | 26.97 | Argillaceous cementation, small cross bedding |
| | Medium sandstone | 15.77 | Calcareous cementation, massive bedding, vertical fracture development |
| | Siltstone | 13.15 | Dark gray, argillaceous cementation, wavy bedding |
| | Fine sandstone | 11.85 | Argillaceous cementation, with small cross bedding and developed fractures |
| | Coal | 12.11 | It is dominated by bright coal and dark coal with developed fractures |
| | Fine sandstone | 23.33 | Argillaceous cementation, with small cross bedding and developed fractures |

**Figure 1.** The position of sandstone sample taken in the test.

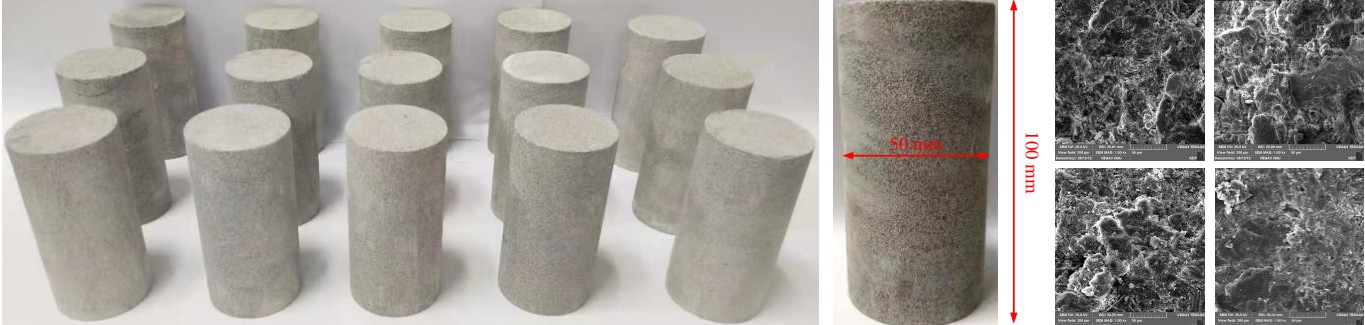

**Figure 2.** Microstructure of sandstone samples and size.

### 2.2. Test System and Principle

Figure 3 presents the schematic diagram of the rock series triaxial permeability experiment system produced by Top Industries. The experimental system was mainly composed of the axial loading system, the confining pressure system, the permeability system, and axial and radial displacement sensors. The core components of the multi-field coupling seepage experiment bench were an oil pump, water pump, and a triaxial pressure chamber. The principle of the permeability experiment was as follows: first, the sample was placed in the cylinder of the triaxial pressure chamber, and axial stress was applied to the sample by the axial loading system ($P_1$ pump), with a maximum axial stress of 500 MPa; then, hydraulic oil was injected into the triaxial pressure chamber, and the confining pressure ($P_2$ pump) was applied to the sample, with a maximum confining pressure of 60 MPa; finally, osmotic pressure was applied to the upper and lower ends of the sample, with a maximum water pressure ($P_3$ pump and $P_4$ pump) of 50 MPa. In order to accurately measure the axial deformation and radial strain, two LVDT displacement sensors were placed in parallel on both sides of the sample with a range of 12 mm and an accuracy of 0.001 mm.

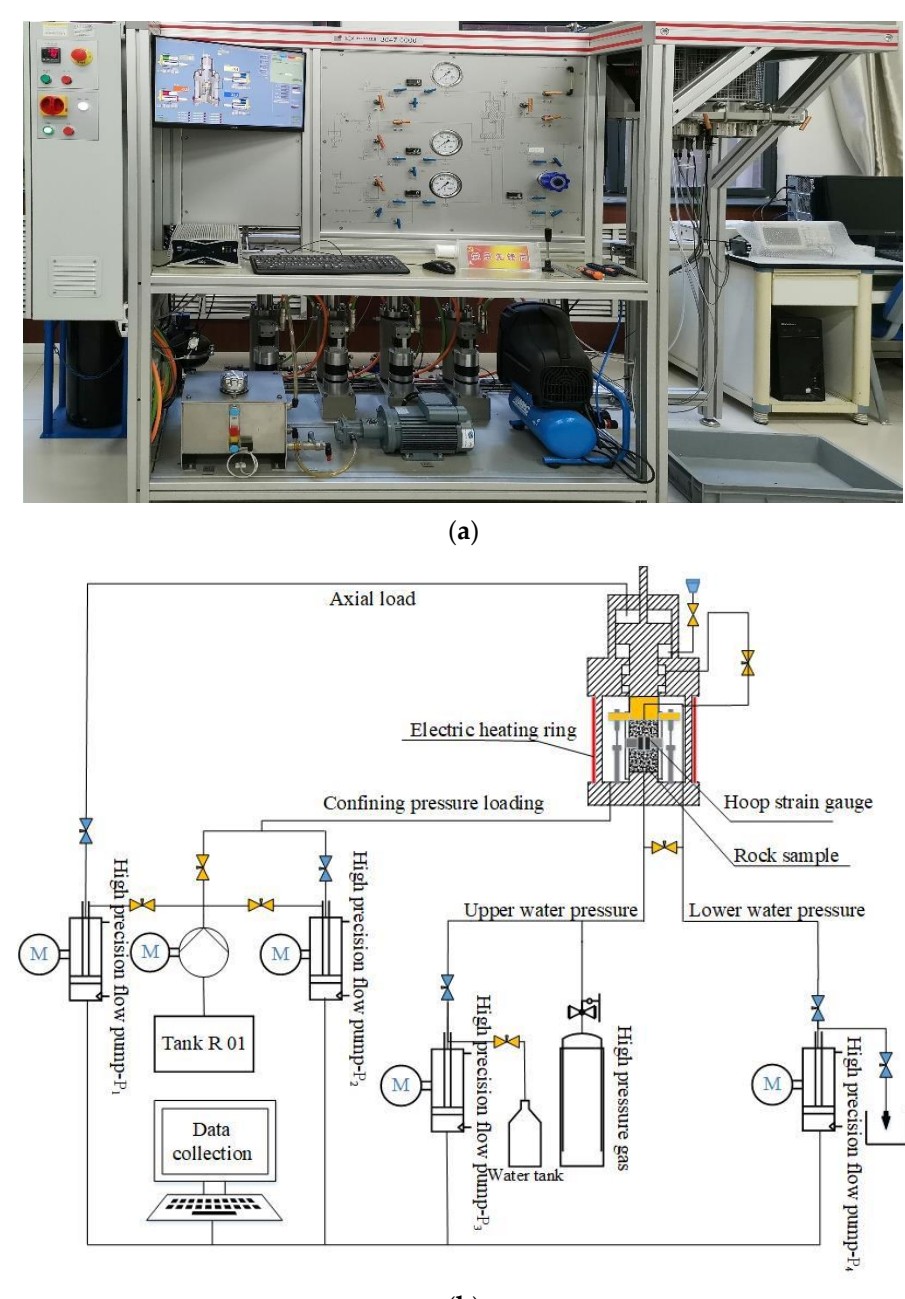

**(a)**

**(b)**

**Figure 3.** Physical and structure schematic diagram of penetration test system. (**a**) Physical drawing. (**b**) Schematic diagram.

### 2.3. Principle of Calculating Sandstone Permeability Based on Non-Darcy's Law

It is well known that for steady-state Darcy flow, there is only one permeability characteristic parameter of rock, namely permeability $k$. For non-Darcy flow, its permeability can be described by three parameters, namely, rock permeability $k$, non-Darcy flow $\beta$ factor, and acceleration coefficient $c_a$. Rocks generally follow non-Darcy flow under the influence of complex mining stress. Indoor determination of rock permeability mainly includes two types: the steady-state permeability method and the transient permeability method. The steady-state method is to obtain the permeability characteristic parameter of rocks according to the curve fitting of the pressure gradient and seepage velocity scatter plot, which is suitable for rocks with high permeability. The transient method is used to collect the pore pressure difference sequence at both ends of the rock sample over a period of time. The time series of seepage velocity and its change rate are obtained by difference. The

permeability characteristic parameters of rock unsteady seepage are calculated based on four time series of the pressure gradient and its change rate, as well as the seepage velocity and its change rate, which is suitable for rock with low permeability [34]. According to the mercury intrusion experimental results, the initial permeability of the sample was about $10^{-17}$ m$^2$, and the permeability is low. Therefore, the transient method was used to determine the permeability parameters of the samples.

The principle of the transient method is shown in Figure 4. The volume of the two tanks in Figure 4 is $B$, the pressures are $P_1$ and $P_2$, and the heights and cross-sectional areas of the specimens are $H$ and $A$, respectively. The pressure at both ends of the specimen at the initial moment is $P_{10}$ and $P_{20}$ ($P_{10} > P_{20}$), respectively. The pressure gradient in the axial direction of the specimen is $(P_{20} - P_{10})/H$. During the permeation process, the liquid in water tank 1 passes through the specimen into water tank 2. The pressure ($P_1$) in tank 1 decreases with the increase of pressure ($P_2$) in tank 2. The pressure gradient $(P_2 - P_1)/H$ gradually decreases until the pressures of the two tanks are equal and reach equilibrium state, as shown in Figure 5.

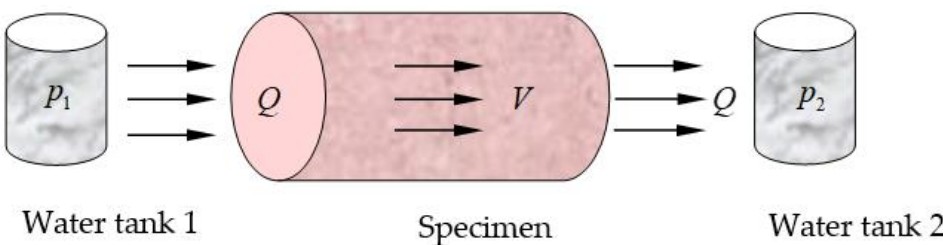

Water tank 1                              Specimen                              Water tank 2

**Figure 4.** Principle of transient penetration test.

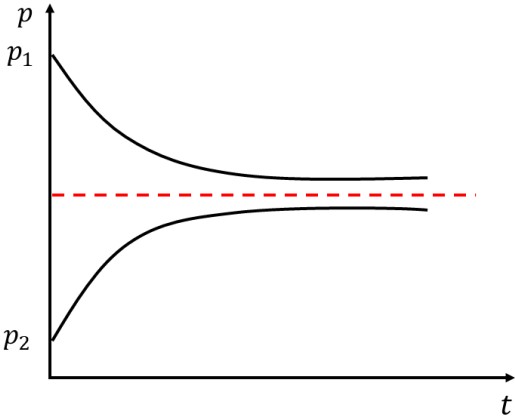

**Figure 5.** The pressure change curve of tank 1 and 2 with time.

Assume that the mass flow rate of water entering the specimen from tank 1 is $Q$. If the specimen is saturated, the mass flow rate of the liquid entering tank 2 is also $Q$. The seepage velocity ($V$) is $V = Q/(\rho A)$. Combining equations $c_f = \frac{1}{\rho}\frac{d\rho}{dp}$ and $d\rho = -\frac{Qdt}{B}$, we can obtain

$$\frac{dP_1}{dt} = -\frac{AV}{c_f B} \tag{1}$$

where, $\rho$ and $c_f$ is the fluid density and coefficient of volume compressibility, respectively. By the same token, we can also obtain

$$\frac{dP_2}{dt} = \frac{AV}{c_f B} \tag{2}$$

Combining Equations (1) and (2), the following equation can be deduced

$$V = \frac{c_f B}{2A} \frac{d(P_1 - P_2)}{dt} \qquad (3)$$

or

$$V = \frac{c_f BH}{2A} \frac{d\xi}{dt} \qquad (4)$$

where, $\xi$ is the pressure gradient. Combining Darcy's law and Equation (3), it is easy to get

$$\frac{d(P_1 - P_2)}{P_1 - P_2} = -2\frac{Ak_D}{c_f BH\mu} dt \qquad (5)$$

where, $\mu$ and $k_D$ are the viscosity coefficient and permeability coefficient, respectively.

Assuming that the experimental data are collected at equal intervals $\tau$, the total number of samples collected is $n$. The tank pressures at the end of time ($t_f = n\tau$) are $P_{1f}$ and $P_{2f}$, respectively. The sequence of the tank pressure difference in the test is $\Delta P = (P_2 - P_1)_{t = i\tau}$, $i = 1, 2, \ldots n$. In this way, the pressure gradient sequence can be calculated as follows $\xi = \Delta P_i / H$, $i = 1, 2, \ldots n$.

Integrating Equation (3), the expression of the Darcy flow permeability of the sample is obtained as

$$k_D = \frac{c_f BH\mu}{2t_f A} \ln \frac{P_{10} - P_{20}}{P_{1f} - P_{2f}} \qquad (6)$$

According to Equation (4), the rate of change of seepage velocity with respect to time can be obtained as

$$\eta = \frac{dV}{dt} = \frac{c_f BH}{2A} \frac{d^2\xi}{dt^2} \qquad (7)$$

On differentiation between Equations (4) and (7), the time series of seepage velocity and the rate of change of seepage velocity with time can be obtained as

$$V_i = \frac{c_f BH}{2A} \frac{\xi_{i+1} - \xi_i}{2\tau}, \ (i = 1, 2, \cdots, n-1) \qquad (8)$$

$$\eta_i = \frac{c_f BH}{2A} \frac{\xi_{i+2} - 2\xi_i + \xi_{i-2}}{4\tau^2}, \ (i = 2, \cdots, n-2) \qquad (9)$$

The conservation of momentum equation for unsteady non-Darcy flow is

$$\rho c_a \eta_i = -\xi_i - \frac{\mu}{k} V_i + \rho \beta V_i^2, \ (i = 2, \cdots, n-2) \qquad (10)$$

According to formula (10), the functionals are constructed as follows:

$$\Pi = \sum_{i=2}^{n-2} \left( -\xi_i - \frac{\mu}{k} V_i + \rho \beta V_i^2 - \rho c_a \eta_i \right)^2 \qquad (11)$$

Equation (11) was used to take derivatives of different parameters. Then, the extreme conditions of functionals are:

$$\frac{\partial \Pi}{\partial (\rho \beta)} = 0, \left( \sum_{i=2}^{N-2} V_i^4 \right) \rho \beta - \left( \sum_{i=2}^{N-2} V_i^3 \right) \frac{\mu}{k} - \left( \sum_{i=2}^{N-2} \eta_i V_i^2 \right) \rho c_a - \left( \sum_{i=2}^{N-2} \xi_i V_i^2 \right) = 0 \qquad (12)$$

$$\frac{\partial \Pi}{\partial (\frac{\mu}{k})} = 0, \left( \sum_{i=2}^{N-2} V_i^3 \right) \rho \beta - \left( \sum_{i=2}^{N-2} V_i^2 \right) \frac{\mu}{k} - \left( \sum_{i=2}^{N-2} \eta_i V_i \right) \rho c_a - \sum_{i=2}^{N-2} \xi_i V_i = 0 \qquad (13)$$

$$\frac{\partial \Pi}{\partial (\rho c_a)} = 0, \left( \sum_{i=2}^{N-2} V_i^2 \eta_i \right) \rho \beta - \left( \sum_{i=2}^{N-2} V_i \eta_i \right) \frac{\mu}{k} - \left( \sum_{i=2}^{N-2} \eta_i^2 \right) \rho c_a - \left( \sum_{i=2}^{N-2} \xi_i \eta_i \right) = 0 \qquad (14)$$

Combining Equations (12)–(14), the characteristic parameters ($k$, $\beta$, $c_a$) of non-Darcy flow can be obtained. Since this paper mainly focuses on the study of specimen permeability evolution law under the influence of CLT and confining pressure, the influence of parameters $\beta$ and $c_a$ will be omitted later.

*2.4. Test Scheme and Procedure*

In order to effectively reflect the post-peak permeability characteristics of sandstone in roof aquifer of 2-2# coal seam, after sample processing and screening, the permeability evolution experiment of sandstone under the post-peak cyclic loading–unloading under different confining pressures was carried out. The temperature of the test environment was maintained at 20 °C and the specific experimental steps were as follows:

(1) The selected rock samples were placed completely in a glass tank full of water before the experiment. They were weighed every half hour and the moisture content was measured until all the samples reached saturation; then the samples were taken out and sealed with preservative film and placed in a cool place to prevent water evaporation. The saturated moisture content of sandstone samples used in this experiment was in the range of 6.54–6.76%, and the error was small, which basically met the experimental requirements.

(2) The specimen was placed in a triaxial pressure chamber, fitted with axial and radial displacement transducers, and axial and radial displacement sensors were installed. Finally, the seal of the seepage channel was checked.

(3) The stress control mode was adopted to apply the design confining pressure to the sample at a loading rate of 0.5 kN/s, and then the axial strain control mode was adopted to axially load the sample at a loading rate of 0.003 mm/s until it reached 75–95% of the peak stress in the post-peak state (as shown in Figure 6: point A*). During the loading period, the permeability point was arranged with a gradient of 0.003 and the permeability was measured. The reason why 75–95% of the peak stress (point A*) was taken as the first unloading point of the post-peak sample was mainly because the sample had strong brittleness and the post-peak curve dropped significantly. Once the stress at the unloading point became too low, the post-peak stress state would be difficult to maintain, which was not conducive to the experiment. In addition, the closer the starting pressure of the stress–strain hysteresis was to the peak value, the more comprehensive the permeability change of the sample under plastic flow would be, and the experimental accuracy would be effectively improved.

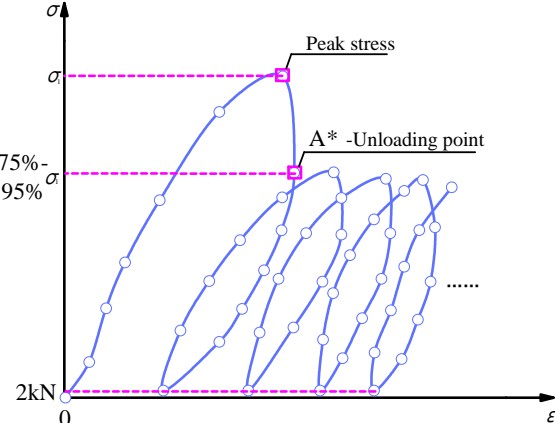

**Figure 6.** Schematic diagram of penetration test point layout.

(4) When the post-peak stress state reached point A*, the loading mode was switched from the axial strain control mode to the axial stress control mode, and the sample was unloaded step by step at a rate of 0.5 kN/s. During the unloading process, the permeability test points were arranged with 10 KN as the stress gradient, until it was reduced to 2 kN, and the unloading was stopped. Subsequently, reverse loading was started, and the

loading rate, the arrangement of permeability points, and the control mode were the same as those of the unloading scheme until the stress state of A* was loaded. Then the cyclic loading–unloading permeability experiment was completed (see Figure 6).

(5) Other cycles were carried out with the same loading–unloading method, so as to complete the cyclic loading–unloading permeability experiment under 4–6 hysteretic curves. The schematic diagram of the penetration test point layout is shown in Figure 7. The confining pressure remains constant throughout the loading and unloading process.

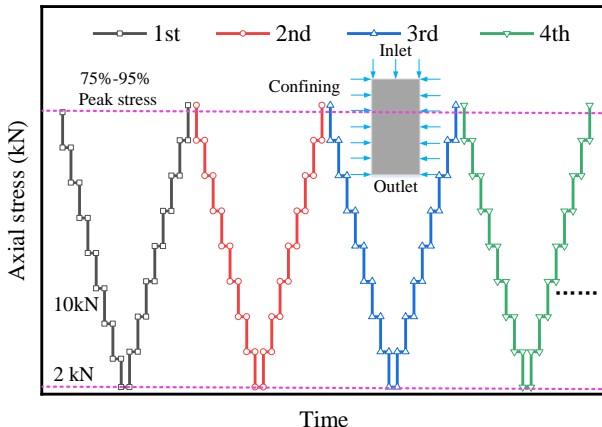

**Figure 7.** Schematic diagram of post-peak penetration test scheme.

(6) To reveal the influence of stress level and post-peak CLT on the permeability characteristics of roof strata, the confining pressures in the test process were set as 4 MPa, 8 MPa, 12 MPa, 16 MPa, and 20 MPa respectively, and the post-peak CLTs were 4–6 times. The pore pressure difference between the upper and lower ends of the sandstone corresponded to the permeability experiment under each confining pressure. The application of the permeability pressure at the upper and lower ends of the sandstone under different confining pressures is shown in Table 1. Operation steps (2) to (5) were repeated for different perimeter pressure levels.

**Table 1.** The confining pressure and corresponding pore water pressure in the test scheme.

| Confining Pressure (MPa) | Hydraulic Pressure (MPa) | |
|---|---|---|
| | Upper Water Pressure | Lower Water Pressure |
| 4 | 3 | 1.5 |
| 8 | 7 | 5.5 |
| 12 | 11 | 9.5 |
| 16 | 15 | 13.5 |
| 20 | 19 | 17.5 |

## 3. Experiment Results and Analysis

### 3.1. Peak Strength and Failure Morphology

The average peak strength of specimens under different confining pressures are 30.56 MPa for $\sigma_3 = 4$ MPa, 57.04 MPa for $\sigma_3 = 8$ MPa, 62.13 MPa for $\sigma_3 = 12$ MPa, 63.25 MPa for $\sigma_3 = 16$ MPa, and 64.68 MPa for $\sigma_3 = 20$ MPa. In addition, Figure 8 shows the typical failure morphology of specimens under various confining pressures. It can be seen from Figure 8 that a main crack runs through the upper and lower ends of the specimen, which is the main seepage channel. With the increase of the confining pressure, the main crack in the upper surface gradually shifts to the bottom, which may have a certain effect on the permeability test results. Shear failure is always the main failure mode no matter whatever the value of the confining pressure. The above analysis is similar to the results of previous infiltration tests.

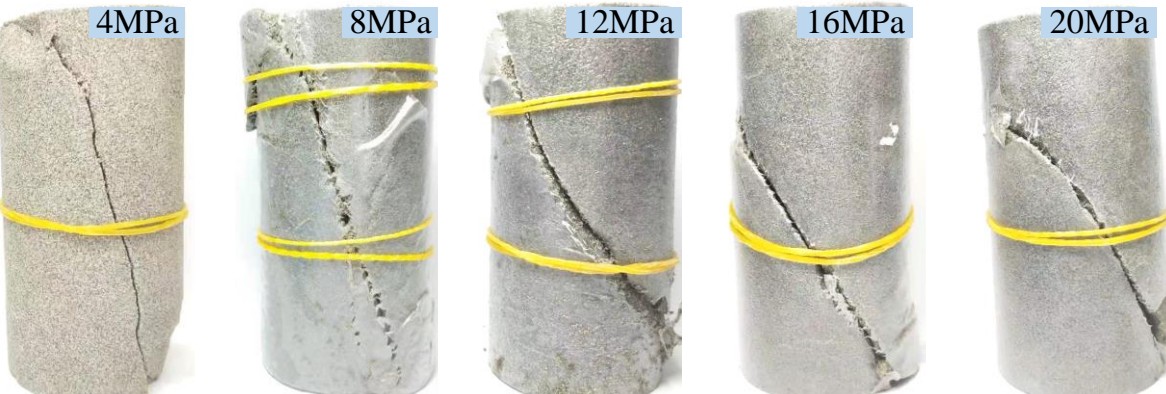

**Figure 8.** Typical failure morphology of specimens under various confining pressures.

*3.2. Influence of Volumetric Strain on Permeability*

The volume strain can directly reflect the volume change of the rock mass, which is a macroscopic index revealing rock fracture behavior and a comprehensive reflection of the influence of axial strain and circumferential strain on permeability. Figure 9 shows the curves of permeability and average stress with volumetric strain under different confining pressures. When the specimens begin to enter the post-peak stress state, the permeability test of 4–6 times cyclic loading–unloading was carried out under different confining pressures and the permeability values for different monitoring sites were also obtained and recorded. Because of the complexity of the test data, the test results at $\sigma_3$ = 4 MPa were taken as examples for analysis. Moreover, the permeability at the inflection point of the stress state was taken as the analysis object. Table 2 shows axial strain, circumferential strain, volumetric strain, and the corresponding permeability at $\sigma_3$ = 4 MPa. The following can be seen from Figure 9 and Table 2:

1. The volume strain has an extremely significant effect on the permeability before the initial unloading point. No matter what the confining pressure was, the pre-peak permeability experienced five stages approximately with the variation of volume strain, an orderly including a slightly initial reduction stage, a slowly increasing stage, a rapid growth stage, an instantaneous drop stage, and a strain softening stage. The above five stages correspond closely to the characteristics of deformation and compression of the specimen before the initial unloading point. During the initial loading phase, the volumetric strain value of the specimen gradually decreased with the compression of the initial crack inside the specimen, which led to a slight decrease in permeability. As the deformation entered the linear elastic stage, the volumetric strain value of the sample started to increase gradually with the development of a new crack, which caused a slow increase of permeability. Next, the new fractures inside the sandstone became expanded and connected with the original fractures as the volumetric strain increased, resulting in a greater rate of permeability growth. Then, once the compressive and deformation state of the sample was close to the peak point, large penetrating cracks were formed inside the sample, and the permeability suddenly increased to the maximum value which lagged the peak stress due to the rapid decrease of the deformation resistance after the peak stress. Meanwhile, the permeability of the specimen entered the instantaneous drop stage when the seepage pressure became stable. Finally, the sample fractures after failure closed again and the permeability decreased as the volumetric strain decreased. At this point, the permeability of the specimen began to enter the strain softening stage. The above analysis belongs to the permeability process for the whole stress–strain stage and many scholars have reached similar conclusions.

2. In the post-peak cyclic loading–unloading stage, the volumetric strain is negatively correlated with permeability. At the unloading stage, when the axial strain increment

was less than that of two times the circumferential strain, the volumetric strain showed an increasing trend. Since the confining pressure remained unchanged at this stage, the permeability decreased because the axial unloading was equivalent to the sample compression. At the initial loading stage, the axial strain increment was also less than that of two times the circumferential strain, the volume strain gradually increased and the sample was compacted, which led to a permeability decrease. Subsequently, the axial strain increment exceeded two times the circumferential strain increment, the volumetric strain curve shifted to the left and showed a decreasing trend. The main fracture across the sample was re-opened, resulting in an obviously increase in permeability. The above analysis showed that the post-peak cyclic loading–unloading process had a significant impact on its permeability, so the change of roof seepage or water inrush after mining needs to take the stress state into account.

**Table 2.** The strain and permeability corresponding to the inflection point of stress state at 4 MPa confining pressure.

| Number | Axial Strain $\varepsilon_1$ (%) | Hoop Strain $\varepsilon_3$ (%) | Volume Strain $\varepsilon_V$ (%) | Permeability $k$ ($10^{-16}$ m$^2$) |
|---|---|---|---|---|
| 1 | 0.0000225 | −0.00527 | −0.01054 | 0.91 |
| 2 | 0.000657 | −0.00533 | −0.01001 | 0.288 |
| 3 | 0.00425 | −0.0091 | −0.01402 | 2.91 |
| 4 | 0.00629 | −0.00939 | −0.01127 | 2.04 |
| 5 | 0.00747 | −0.0113 | −0.01521 | 4.41 |
| 6 | 0.00476 | −0.00851 | −0.01224 | 2.83 |
| 7 | 0.00824 | −0.0123 | −0.01628 | 5.22 |
| 8 | 0.00559 | −0.00947 | −0.01335 | 4.01 |
| 9 | 0.00966 | −0.0136 | −0.01755 | 5.72 |
| 10 | 0.00667 | −0.0106 | −0.0145 | 4.88 |
| 11 | 0.0101 | −0.0145 | −0.01894 | 5.85 |
| 12 | 0.005691 | −0.01139 | −0.01708 | 4.94 |

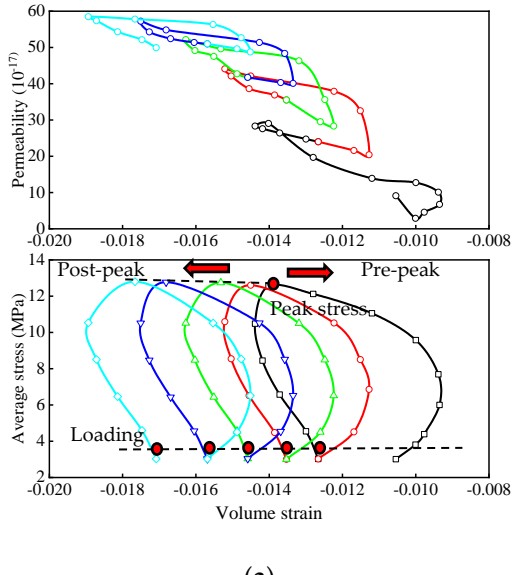

(**a**)

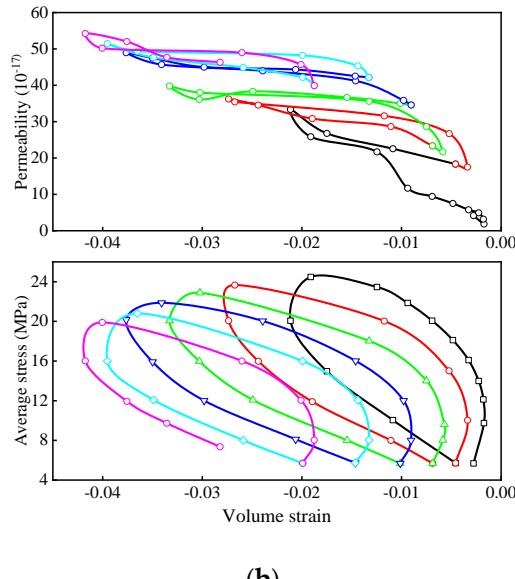

(**b**)

**Figure 9.** *Cont.*

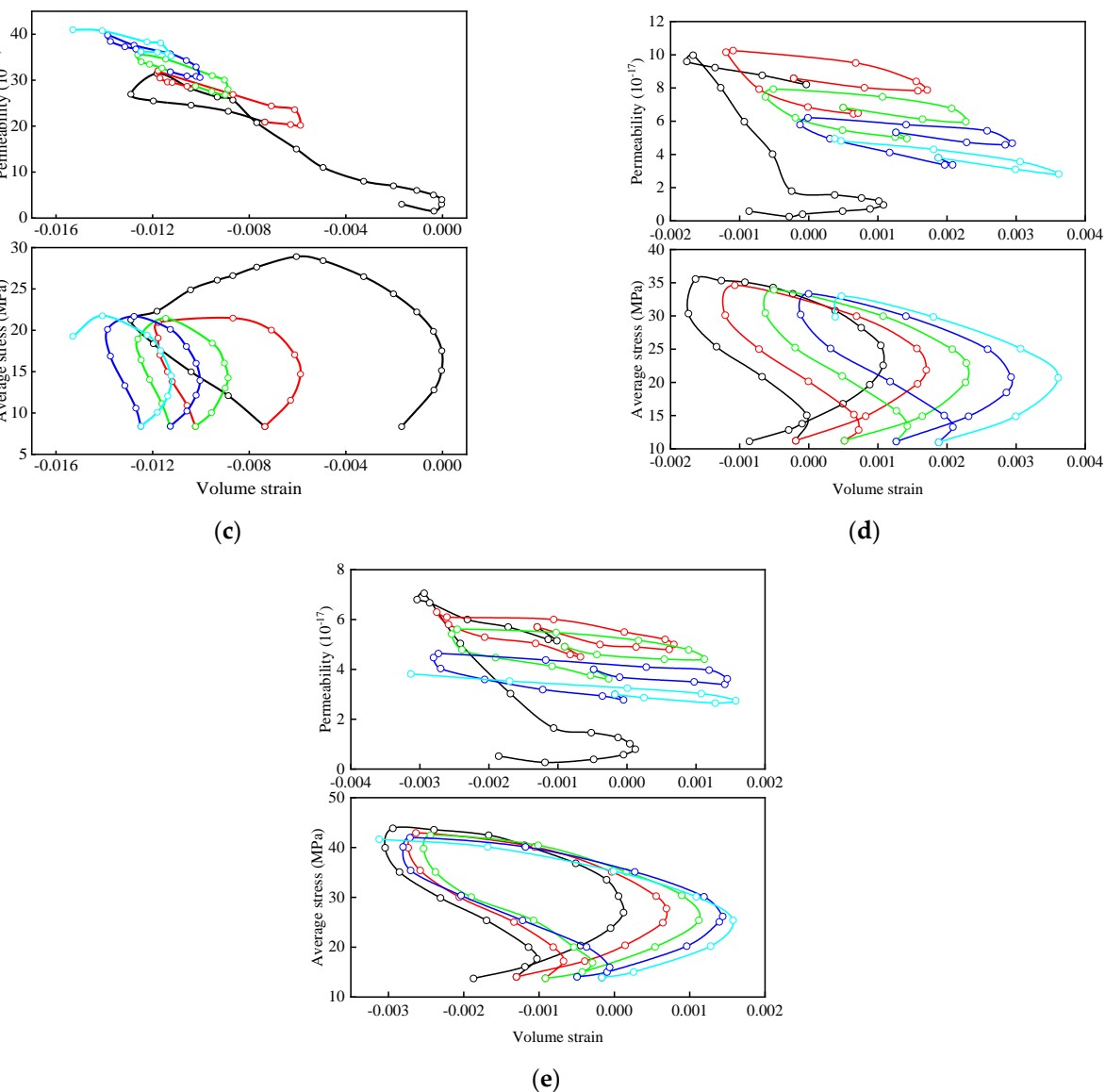

**Figure 9.** Relation curve of permeability and average stress with volumetric strain under different confining pressures. (**a**) $\sigma_3 = 4$ MPa. (**b**) $\sigma_3 = 8$ MPa. (**c**) $\sigma_3 = 12$ MPa. (**d**) $\sigma_3 = 16$ MPa. (**e**) $\sigma_3 = 20$ MPa.

### 3.3. Influence of Confining Pressure and CLT on Permeability

Figure 10 shows the variation of permeability with confining pressure and CLT, respectively. Table 3 presents the permeability value of various monitoring points under different confining pressures and CLT. The following can be summarized from Figure 10 and Table 3:

1.  The influence of CLT on *k* is closely related to the confining pressure level. When the confining pressure changed from 4 MPa to 12 MPa, the circumferential strain gradually decreased as the axial strain increased and the final volumetric strain also decreased, which caused the volumetric strain–average stress hysteretic curve to shift to the left in turn and the corresponding permeability gradually increased. When the confining pressure increased to 16 MPa and 20 MPa, the influence of the confining pressure on the volumetric strain was greater than that of the axial stress as CLT increased. The volumetric strain–average stress hysteretic curve shifted to the right in turn and the corresponding permeability showed a decreasing trend.

2.  The confining pressure has an extremely significant effect on the post-peak permeability of specimens. No matter what the value of CLT, the magnitude of the sandstone permeability gradually decreased and the decreasing trend became flat as the confin-

ing pressure increased, especially for $\sigma_3$ = 16 MPa and 20 MPa. This was due to the larger confining pressure, the denser pores and the cracks which were compressed. The water flow channel in the rock medium contained cracks and pores, the stress surface of the pore part was arched, and the stress was compressive stress, therefore the pore volume did not change much, resulting in a flat decreasing trend. Compared with pores, cracks were more prone to compression deformation and closure due to the lack of rock skeleton matrix support. Therefore, the whole seepage process was first fracture compression deformation and closure, and then extrusion deformation of some pores. Under high confining pressures, the deformation and closure of flow channels such as fractures and effective pores were more and more difficult to achieve as the confining pressure increased. Therefore, the higher the confining pressure, the smoother was the permeability variation.

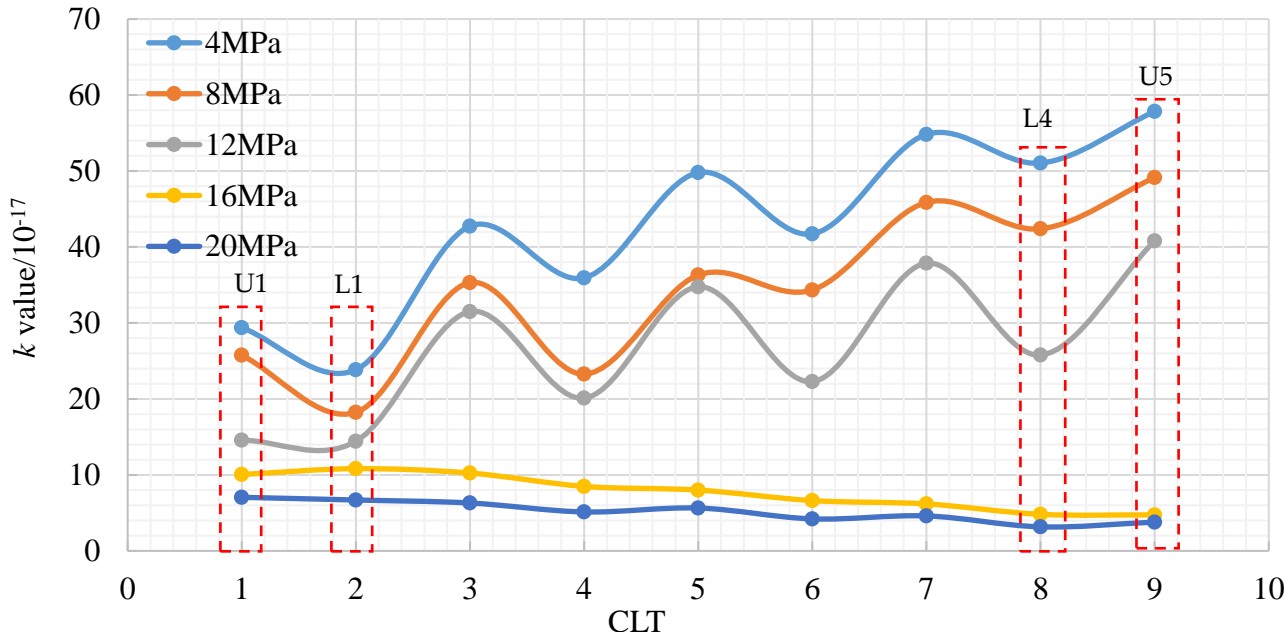

**Figure 10.** Variation law of permeability of different monitoring points with confining pressure and CLT.

**Table 3.** $k$ value of various monitoring points under different confining pressures and CLT.

|  | $\sigma_3$/MPa | $k/10^{-17}$ | | | | |
|---|---|---|---|---|---|---|
|  |  | U1 | U2 | U3 | U4 | U5 |
| Unloading | 4 | 29.383 | 42.741 | 49.792 | 54.820 | 57.837 |
|  | 8 | 25.755 | 35.307 | 36.305 | 45.866 | 49.134 |
|  | 12 | 14.580 | 31.493 | 34.761 | 37.869 | 40.804 |
|  | 16 | 10.061 | 10.247 | 8.007 | 6.187 | 4.740 |
|  | 20 | 7.056 | 6.308 | 5.644 | 4.608 | 3.790 |
|  | $\sigma_3$/MPa | $k/10^{-17}$ | | | | |
|  |  | L1 | L2 | L3 | L4 | L5 |
| Loading | 4 | 23.850 | 35.943 | 41.731 | 51.049 | 49.523 |
|  | 8 | 18.235 | 23.267 | 34.343 | 42.393 | 40.364 |
|  | 12 | 14.432 | 20.111 | 22.283 | 25.785 | - |
|  | 16 | 10.825 | 8.491 | 6.624 | 4.816 | - |
|  | 20 | 6.699 | 5.130 | 4.215 | 3.169 | - |

*3.4. Influence of CLT on Recovery Rate of Permeability*

Figure 11 presents a typical volumetric strain–average stress hysteresis curve, which can be used to describe the area variation of the sandstone hysteresis curve by defining the width–length ratio. The ratio value can reflect the size of plastic deformation and the recovery rate of the permeability of sandstone [12]. In Figure 11, the coordinates of point $A$ and point $B$ were $(X_1, Y_1)$ and $(X_2, Y_2)$, respectively. The coordinates of point $C$ and point $D$ were $(X_3, Y_3)$ and $(X_4, Y_4)$, respectively. The length and width of the hysteresis curve were $l$ and $d$, respectively, so the expression of the hysteresis width–length ratio was:

$$\eta = \frac{d}{l} = \frac{\sqrt{(X_4 - X_3)^2 + (Y_4 - Y_3)^2}}{\sqrt{(Y_2 - Y_1)^2 + (X_2 - X_1)^2}} \tag{15}$$

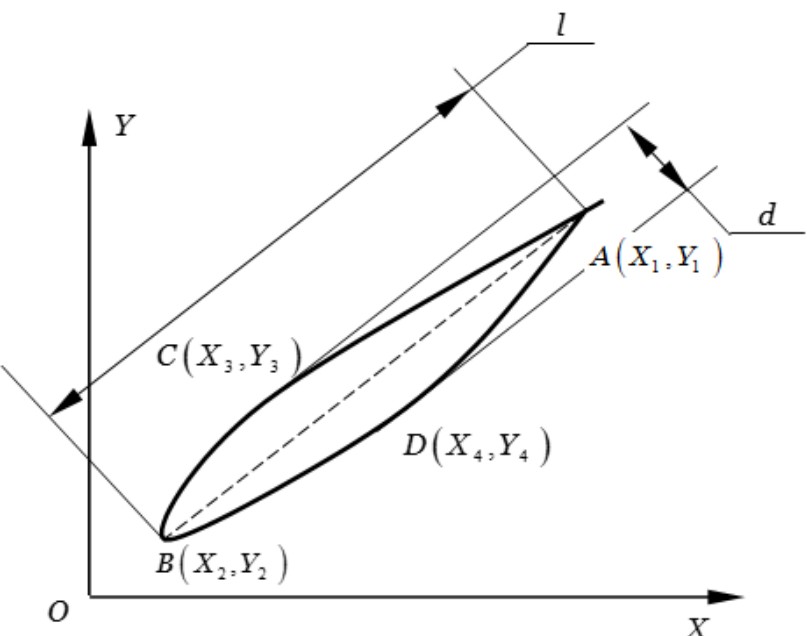

**Figure 11.** Schematic diagram of breadth–length ratio characterization methods of typical volumetric strain–average stress hysteretic curves.

The breadth–length ratio values of the volumetric strain–average stress hysteretic curve under different CLT and confining pressures are shown in Table 4. It can be seen from Table 4 that no matter what the value of the confining pressure, the hysteresis area of the first cycle was larger than that of the last three cycles. The hysteresis areas of the last three cycles are basically equal, indicating that the plastic deformation generated in the first cycle was larger than that generated in the last three cycles. In addition, the corresponding permeability curve does not form a closed loop during loading and unloading. In first loading–unloading process, the non-closed hysteresis area formed by the volumetric strain-permeability was larger, and the area of the latter three hysteresis curves was smaller (see Figure 9), which was similar to the results of the volumetric strain–average stress hysteresis. This indicated the sample mainly produced elastic deformation after plastic deformation and the recovery rate of the permeability increased with the increase of the cycle times.

**Table 4.** Breadth–length ratio values of volumetric strain–average stress hysteretic curve.

| Confining Pressure (MPa) | Cyclic Loading–Unloading Time | | | |
|:---:|:---:|:---:|:---:|:---:|
| | 1 | 2 | 3 | 4 |
| 4 | 0.00134 | 0.00127 | 0.00122 | 0.00123 |
| 8 | 0.00269 | 0.00242 | 0.00251 | 0.00248 |
| 12 | 0.0049 | 0.0035 | 0.0032 | 0.0037 |
| 16 | 0.0065 | 0.0051 | 0.0057 | 0.0054 |
| 20 | 0.0076 | 0.0061 | 0.0059 | 0.0064 |

## 4. Microscopic Mechanism Analysis

In order to reveal the permeability variation with the confining pressure with the microscopic mechanism, some initial and post-experiment samples were selected to carry out the SEM scanning test. Then, based on the 2D SEM image with high resolution, a numerical reconstruction calculation was carried out by using the random reconstruction method to obtain the 3D micro-structure characteristics of samples before and after the experiment, and the 3D pore model was characterized. The characterization parameters generally included pore diameter, effective porosity, connectivity, seepage path length, and tortuosity [33].

### 4.1. D Reconstruction Principle

The 3D model reconstruction of the sandstone internal structure was carried out based on high-resolution microscopic images obtained by SEM scanning experiments. In this paper, the optimized simulated annealing method was used for reconstruction. The reconstruction process was mainly divided into four parts as follows: (1) Preprocessing, segmentation. and initial reconstruction of 2D SEM images; (2) Establishment of statistical function and calculation of model parameters; (3) Calculation and update of the system energy; (4) Final 3D reconstruction of SEM images. The threshold segmentation algorithm and the 3D reconstruction principle of the 2D SEM images are mainly introduced below, while the specific calculation process is shown in the literature [33].

(1) The segmentation algorithm of 2D SEM images.

Fractured sandstone, whose gray histogram was generally in bimodal distribution, was used in this paper and therefore, the pixel gradient distribution algorithm was carried out to segment 2D SEM images. The maximum peak in the gray histogram represented the number of rock matrix pixels, and the other smaller peaks represented the number of fracture structure pixels. The main algorithm principle was to calculate the second derivative $a_i$ of the gray histogram curve. The gray level corresponding to the maximum $a_i$ was a reasonable threshold for segmenting the fracture structure image. The specific expression was as follows:

$$a_i = \frac{\partial^2 Q_i}{\partial i^2}, \; i_{Q\text{max}1} < T < i_{Q\text{max}2} \tag{16}$$

where, $i$ was the gray level, $Q_i$ was the corresponding number of pixels, $i_{Q\text{max}1}$ and $i_{Q\text{max}2}$ were the gray level corresponding to the smaller peak and the maximum peak of the gray histogram curve, respectively.

(2) Optimization of Simulated Annealing Method.

The numerical reconstruction algorithm was based on simulated annealing method and optimized in the form of statistical function increment. The specific incremental improvement equation was as follows:

$$\begin{aligned} \Delta C &= C_{new} - C_{ini} \\ &= 2 \times \left(N_{row}^{new} - r\right) + 2 \times \left(N_{column}^{new} - r\right) \\ &\quad -2 \times \left(N_{row}^{ini} - r\right) - 2 \times \left(N_{column}^{ini} - r\right) \end{aligned} \tag{17}$$

where, $C_{ini}$ and $C_{new}$ corresponded to the contribution value of selected pixels (hole, fissure or matrix phase) to the statistical function before and after pixel position exchange, and $r$ was the statistical distance. $N_{row}^{ini}$, $N_{column}^{ini}$, $N_{row}^{new}$ and $N_{column}^{new}$ were the total number of continuous adjacent holes and cracks in row and column directions before and after pixel position exchange, which can be calculated by the following formula:

$$
\begin{aligned}
N_{row}^{ini} &= \begin{cases} N_{1-l} + N_{1-r} + 1, & Pore \\ N_{0-l} \\ N_{0-r} \end{cases}, \qquad Matrix \\[2mm]
N_{column}^{ini} &= \begin{cases} 0, & Pore \\ N_{0-u} \\ N_{0-d} \end{cases}, \quad Matrix \\[2mm]
N_{row}^{new} &= \begin{cases} N_{1-l} \\ N_{1-r} \end{cases}, \qquad Matrix \\ & \quad N_{0-l} + N_{0-r} + 1, \quad Pore \\[2mm]
N_{column}^{new} &= \begin{cases} N_{0-u} + (Row_1 - Row_0) \\ N_{0-d} - (Row_1 - Row_0) \end{cases}, \quad Matrix \\ & \quad 0, \qquad\qquad\qquad\qquad Pore
\end{aligned}
\tag{18}
$$

where, the matrix pixels in four directions were denoted as $N_{0-u}$, $N_{0-d}$, $N_{0-l}$ and $N_{0-r}$, respectively; the hole and fracture pixels in four directions were denoted as $N_{1-u}$, $N_{1-d}$, $N_{1-l}$ and $N_{1-r}$, respectively; $Row_0$ and $Row_1$ were the rows and columns of selected hole and fracture phases, respectively.

*4.2. Characterization Parameters of 3D Pore and Fracture Model*

To quantitatively analyze the variation characteristics of the sample internal pore structure, the 3D pore structure parameters were quantitatively extracted and characterized under different confining pressures, including pore diameter, effective porosity, connectivity, seepage path length, and tortuosity. The pore diameter is the average of all pore sizes in the continuous pore distribution. The effective porosity is the sum of the interconnected pores. The connectivity reflects the connected degree between pores and fractures. The seepage path length is the fluid flowing distance in the real pore structure, and the tortuosity reflects the tortuous degree of seepage path; both parameters take the values corresponding to the highest frequency distribution. The above parameters can directly reflect the permeability variation under different loading conditions. The specific calculation methods of the above parameters are not described in this paper, please refer to references [32,33].

*4.3. 3D Reconstruction Results and Characterization of Pore and Fracture Model*

The 3D model reconstructed in this paper includes two parts: matrix and pore. The connected pore network model is the main fluid flow channel. In order to compare effectively the differences in internal seepage paths of the sample under different confining pressures, it was necessary to extract the pore network and rock matrix model. The 3D internal structure of samples under different confining pressures before and after the experiment are shown in Figure 12.

The characterization parameters of the 3D reconstruction results of the internal structure before and after the experiment under different confining pressures are shown in Table 5. The variation of these parameters with confining pressure is shown in Figure 13. The following can be seen from Table 5 and Figure 13:

1.  The pore diameter, effective porosity, and the connectivity of the initial samples were obviously less than that of the post-experiment samples, and the seepage path length and tortuosity of the initial samples were visibly longer and larger than that of the post-experiment samples, which showed that the pores and cracks in the post-

experimental samples significantly increased, so the permeability obviously increased compared with the initial samples.

2.  The confining pressure level has a significant effect on the seepage parameter of sandstone. As the confining pressure increased, the parameters of pore diameter, effective porosity, and connectivity show an approximately linear decreasing trend while the seepage path length and tortuosity increased nonlinearly. For instance, when $\sigma_3$ changes from 4 MPa to 20 MPa, the pore diameter, effective porosity, and connectivity of the specimen decreased from 9.53 μm to 6.53 μm, from 57.96% to 36.71%, and from 80.35% to 51.29%, respectively. While the seepage path length and tortuosity of the specimen increased from 45.43 μm to 72.69 μm and from 2.9 to 4.8, respectively. The above analysis indicates that with the increase of confining pressure, the shear failure of the specimen becomes more pronounced, and the tortuosity and seepage path become larger and longer, respectively. It means that the pores inside the sample were easier to be compacted, and the fractures formed during failure were easier to be closed, which eventually led to decreasing permeability. Therefore, the rationality of the above permeability variation with confining pressure was verified from a micro-perspective.

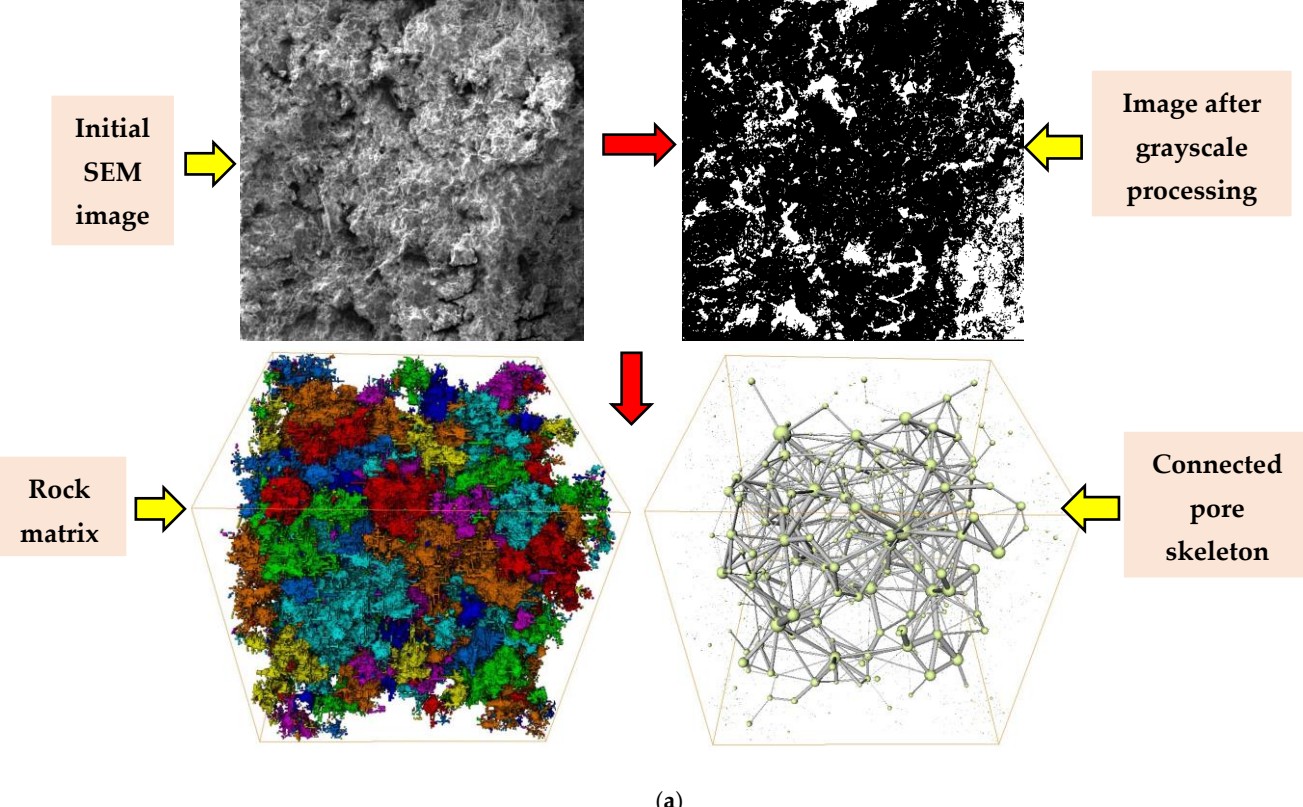

(**a**)

**Figure 12.** *Cont.*

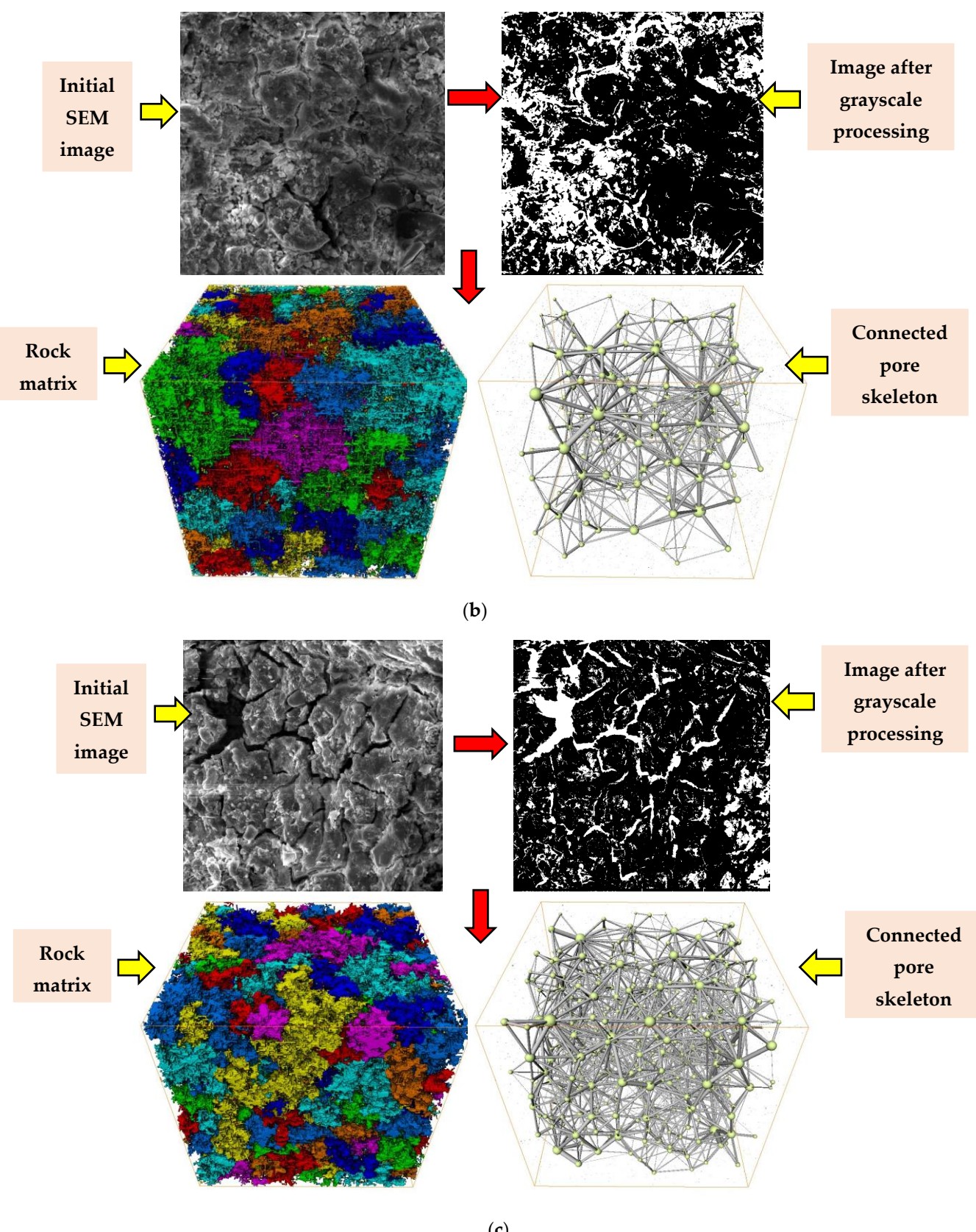

**Figure 12.** *Cont.*

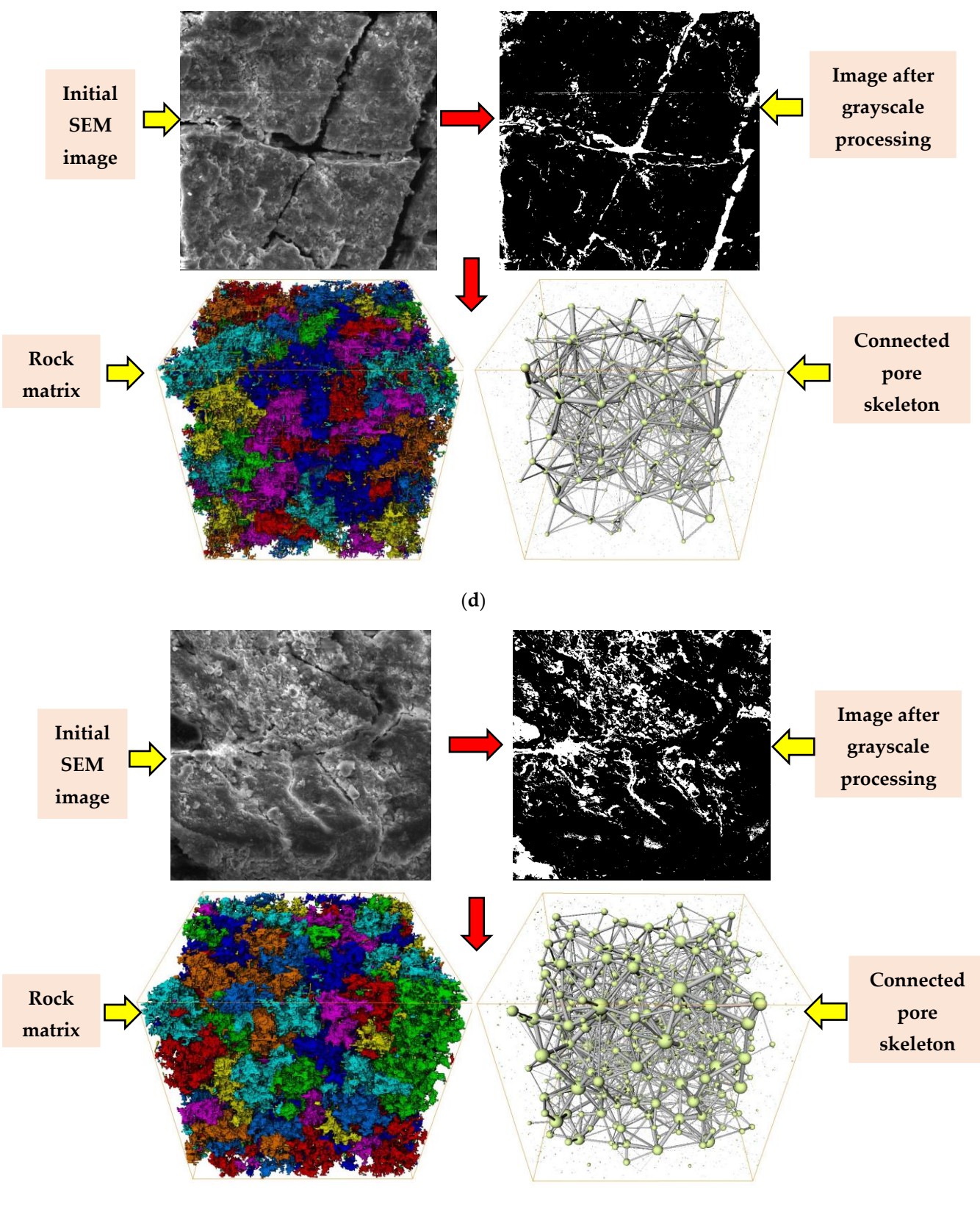

(**d**)

(**e**)

**Figure 12.** *Cont.*

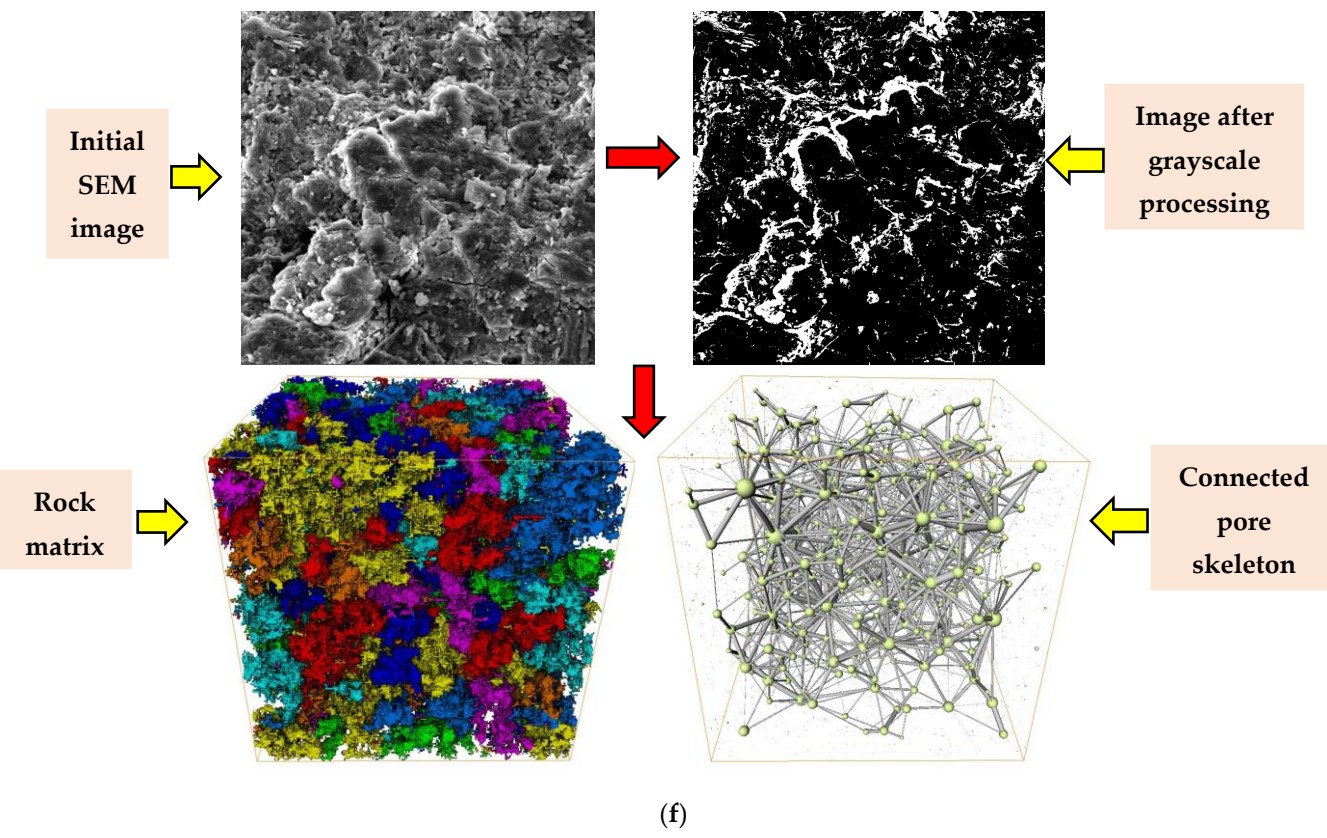

(**f**)

**Figure 12.** 3D reconstruction results of internal structure of sandstone under different confining pressures. (**a**) 3D reconstruction results of internal structure of initial state sandstone. (**b**) $\sigma_3 = 4$ MPa. (**c**) $\sigma_3 = 8$ MPa. (**d**) $\sigma_3 = 12$ MPa. (**e**) $\sigma_3 = 16$ MPa. (**f**) $\sigma_3 = 20$ MPa.

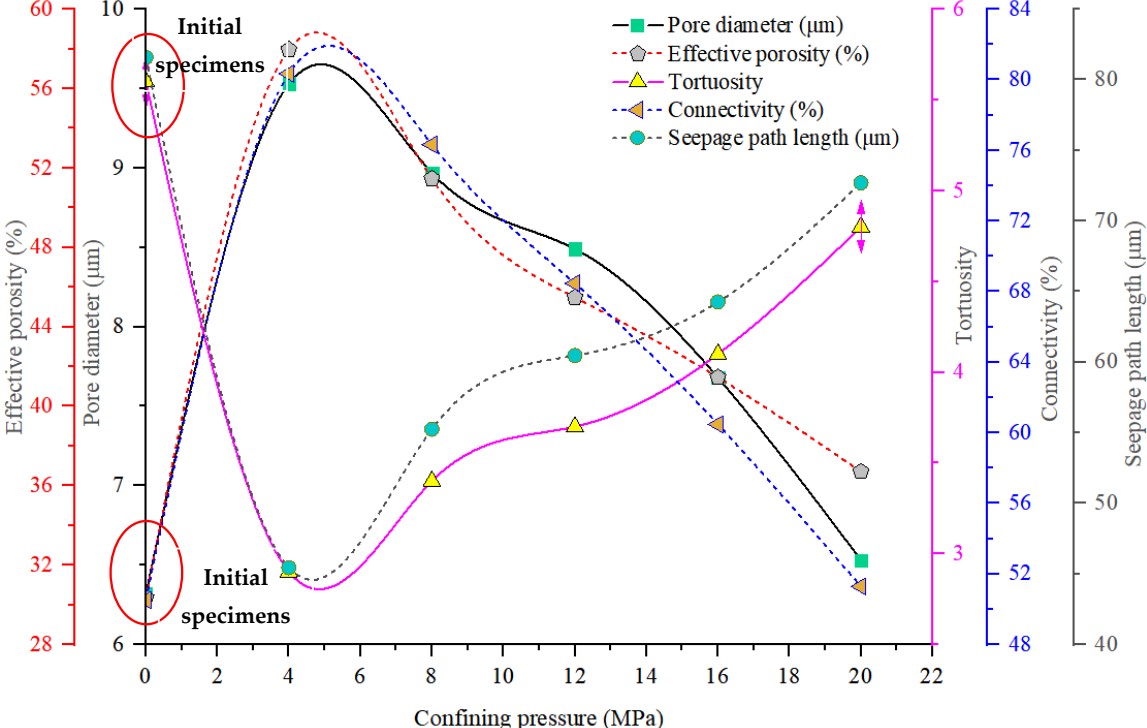

**Figure 13.** The variation law of seepage parameters (pore diameter, effective porosity, connectivity, seepage path length, and tortuosity) with confining pressure before and after test.

**Table 5.** Characterization parameters of 3D internal structure of sandstone before and after test.

| Experimental Condition | Confining Pressure (MPa) | Pore Diameter (μm) | Effective Porosity (%) | Connectivity (%) | Seepage Path Length (μm) | Tortuosity |
|---|---|---|---|---|---|---|
| Before testing | 0 | 6.32 | 30.36 | 50.52 | 81.58 | 5.6 |
| After testing | 4 | 9.53 | 57.96 | 80.35 | 45.43 | 2.9 |
| | 8 | 8.97 | 51.47 | 76.32 | 55.24 | 3.4 |
| | 12 | 8.49 | 45.48 | 68.46 | 60.47 | 3.7 |
| | 16 | 7.68 | 41.46 | 60.47 | 64.25 | 4.1 |
| | 20 | 6.53 | 36.71 | 51.29 | 72.69 | 4.8 |

## 5. Conclusions

In this paper, with the cyclic loading–unloading permeability experiment of saturated sandstone after shear yield, the influences of confining pressure, cyclic loading–unloading times, and the volumetric strain on permeability were studied systematically. Following this, based on an improved simulated annealing algorithm, 3D reconstruction of the internal structure of saturated sandstone samples before and after the experiment was realized from a microscopic point of view. The influence of the confining pressure on the post-peak permeability variation of saturated sandstone was also discussed. The main conclusions are as follows:

(a) Under a certain confining pressure, with the increase of axial strain, the pre-peak permeability experienced approximately five stages with the variation of volume strain, an orderly including a slightly initial reduction stage, a slowly increasing stage, a rapid growth stage, an instantaneous drop stage, and a strain softening stage. In the post-peak cyclic loading–unloading stage, the volumetric strain was negatively correlated with permeability. At the unloading and initial loading stage, the volumetric strain showed a gradually decreasing trend as the specimen was slowly compressed. However, at the middle and final loading stages, the axial strain increment exceeded two times the circumferential strain increment, while the volumetric strain curve shifted to the left and showed a decreasing trend, resulting in an obvious increase in permeability.

(b) The influence of CLT on $k$ is closely related to the confining pressure level. When the confining pressure changed from 4 MPa to 12 MPa, the volumetric strain–average stress hysteretic curve shifted to the left in turn and the corresponding permeability gradually increased. When the confining pressure increased to 16 MPa and 20 MPa, the volumetric strain–average stress hysteretic curve shifted to the right in turn and the corresponding permeability showed a decreasing trend. No matter what the value of CLT, the magnitude of sandstone permeability gradually decreased and the decreasing trend became flat as the confining pressure increased, especially for $\sigma_3$ = 16 MPa and 20 MPa.

(c) A method to characterize the recovery rate of post-peak permeability was proposed by calculating the breadth–length ratio of volumetric strain–average stress hysteretic curve under different CLT and confining pressures. No matter what the value of the confining pressure, the hysteresis area of the first cycle was larger than that of the last three cycles. The hysteresis areas of last three cycles are basically equal, indicating that the plastic deformation generated in the first cycle was larger than that generated in the last three cycles. This indicated that the sample mainly produced elastic deformation after plastic deformation and the recovery rate of the permeability increased with the increase of CLT.

(d) Based on 3D reconstruction results of the sandstone's internal structure before and after the experiment, pore diameter, effective porosity, and connectivity of the initial sample were smaller than those after the experiment, while seepage path length and tortuosity were larger than those after the experiment. As the confining pressure increased, the pore diameter, effective porosity, and connectivity all decreased due

to the more easily compacted pores and cracks under the high confining pressure, lower connectivity, and permeability; but the length and tortuosity of the seepage path increased due to a more significant shear failure phenomenon where the seepage path became more tortuous, that is, the greater the tortuosity, the longer the seepage path.

**Author Contributions:** Conceptualization, D.Z.; methodology, L.C.; software, J.G.; validation, L.C. and P.W.; formal analysis, G.F.; investigation, X.W and S.Z.; resources, W.Z.; data curation, N.Y.; writing—original draft preparation, L.C.; writing—review and editing, P.W.; visualization, G.F.; supervision, X.W.; project administration, D.Z.; funding acquisition, L.C. All authors have read and agreed to the published version of the manuscript.

**Funding:** This research was funded by National Natural Science Foundation of China (52104100, 51804278, 51874277), China Postdoctoral Science Foundation (2021M703503), open-ended fund of Hubei Key Laboratory for Efficient Utilization and Agglomeration of Metallurgic Mineral Resources (2020zy002), and the Independent Research Project of State Key Laboratory of Coal Resources and Safe Mining (No. SKLCRSM2020x04).

**Institutional Review Board Statement:** Not applicable.

**Informed Consent Statement:** Not applicable.

**Data Availability Statement:** All data, models, or codes that support the findings of this study are available from the corresponding author upon reasonable request.

**Acknowledgments:** Reviewers are thanked for their insightful suggestions and comments, which improved the quality of this manuscript.

**Conflicts of Interest:** The authors declare no conflict of interest.

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
