# Peer review of "Experimental Investigation on Post-Peak Permeability Evolution Law of Saturated Sandstone under Various Cyclic Loading–Unloading and Confining Pressure"

_water, doi:10.3390/w14111773_

Round 1
Reviewer 1 Report
Review: Experimental Investigation on Post-Peak Permeability Evolution Law of Saturated Sandstone under Various Cyclic Loading-Unloading and Confining Pressure
Overall the paper needs major revisions. More details of the calculations of parameters relating to non-Darcy flow need to be given. Clarification of the model for non-Darcy flow is needed and how parameters are calculated. A justification of the non-Darcy model for the permeability of the sandstone sample in the post-peak phase is needed. It should be very clear to reader if the permeability measurements at varying confining pressure are for a pre-fractured sample post-peak or an intact sample that has not reached its peak failure strength. For example if macrofractures are present is the non-Darcy flow model presented in Section 2.3 still applicable? Please justify with references. There needs to be more detailed discussion of why the specific experimental tests were chosen and what information each one gives.
Specific comments:
Abstract line 49-50: How do the conditions here compare with conditions of water-rich working face under multiple mining? This sentence needs to be reworded to explain what is meant by a water-rich working face and what is meant by multiple mining.
Lines 78-80: that is, the influence of rock yield and plastic flow on permeability was not considered, which is very important to reveal the water inrush mechanism of overburden roof fracture under mining as shown in this paper?
Figure 3 shows a triaxial load set-up whereas in Figure 4 only confining pressure is shown. It would be good to have a short description outlining why the two tests are needed and what each test measures and if the sandstone specimens have reached their failure strength in the first triaxial test. Section 2.2-2.4 need more details on what each experiment is for and which experimental parameters are measured under which conditions.
Section 2.2 should be renamed to give more details of what Experiment system is. It would be good to include some figures of some of the cylinders after the triaxial experiment where the axial load reaches 500MPa.
What is the scale of SEM images in Figure 10. Can these be compared with photos of the cylinders after the triaxial experiment for the initial figure 10(a)? If macrofractures are present does this change the analysis for non-Darcy flow presented in Section 2.3?
Section 2.2 What is the compressive strength of the cylinders for varying confining pressure? Is the maximum axial stress of 500MPa greater than the failure strength of all the cylinders? In Figure 5 it shows initial loading to peak stress then unloading from 75-95% of the peak stress. If not loaded to the peak stress, it would be expected that there is very little damage in the linear elastic phase of the sandstone before brittle damage takes place. However as the confining pressure in a triaxial experiment the damage and/or reversible microcrack opening decreases (see Fig. 1(a) in Olsen-Kettle (2019) Int. J. Damage Mech., 28:219-232 and experimental results of Scott T, Ma Q and Roegiers JC (1993) Acoustic velocity changes during shear enhanced compaction of sandstone. Int. J. Rock Mech. Min. Sci. & Geomech. Abstr. 30: 763–769.). So it is important to show photos of the sandstone after triaxial loading at different confining pressures or clearly state if the same confining pressure is used for the triaxial stress loading phase and how this changes if the sample is loaded to peak failure or not, and how this confining pressure is changed for the unloading/loading phases.
Section 2.3 should be renamed to give more details of what Test principle means.
Section 2.3: Is the initial permeability of the sample is about 10-17m 2 , and permeability is low for the sandstone before the triaxial loading? Shouldn’t the initial permeability be reported for the postpeak sandstone with confining pressure = 0?
Are only the confining pressure tests analysed in section 4.3?
Line 188: Combining equations (7), (8) and (9) can get the permeability k of non-Darcy flow.
This line needs to be expanded to give more details of how exactly k is calculated from these equations. How are cf, β and cα measured/calculated?
Line 180: is the time interval t or τ as in equations (3) -(4)?
Minor grammatical errors:
Line 81: In fact, the load borne by the rock is the only external condition
Line 127: Finally, the sample density and longitudinal wave velocity were measured,
Line 133: Figure 2 shows
Line 134: It can be seen
Line 150: Indoor? Replace with Laboratory? Determination
Line 171: the transient method is used to experimentally determine
Line 239 Figure 7 shows
Line 332: Figure 9

Reviewer 2 Report
This paper shows an experimental investigation on post-peak permeability evolution law of saturated sandstone under various cyclic loading-unloading and confining pressure. Based on SEM images and improved simulated annealing algorithm, the 3D internal structure characteristics of sandstone samples before and after experiment are also reconstructed. Influences of confining pressure on seepage parameters of sandstone before and after test are also discussed. It is of great significance to reveal the water inrush mechanism of coal mine. Overall, the investigation is systematic and scientifically sound. The language and logic are also good. However, it has the following minor problems to address before publication:
(1) In section 2.1, the author should give reasons for choosing sandstone as the sample target. This is crucial to the study background.
(2) In section 2.3, the explanation of some symbols is missing, please supplement it. For instance, and N et al in Equations (5) and (7).
(3) In Figure 5, the point of 75%-95% peak strength is selected as the unloading points. Does this have a significant impact on test results due to different unloading points? Please explain it.
(4) In Table 3, the symbols (U1 and L1) are not indicated in Figure 8, please add it.
(5) Some grammatical errors in the article need to be revised completely. Please check and correct it.
